# Developing Edible Starch Film Used for Packaging Seasonings in Instant Noodles

**DOI:** 10.3390/foods10123105

**Published:** 2021-12-14

**Authors:** Hui Chen, Mahafooj Alee, Ying Chen, Yinglin Zhou, Mao Yang, Amjad Ali, Hongsheng Liu, Ling Chen, Long Yu

**Affiliations:** 1Collage of Food Science and Engineering, South China University of Technology, Guangzhou 510640, China; 201920125371@mail.scut.edu.cn (H.C.); 201912800038@mail.scut.edu.cn (M.A.); 201620119572@mail.scut.edu.cn (Y.C.); 202020126236@mail.scut.edu.cn (Y.Z.); 202021027102@mail.scut.edu.cn (M.Y.); dr.amjadali@kiu.edu.pk (A.A.); hs.liu@mail.scut.edu.cn (H.L.); felchen@scut.edu.cn (L.C.); 2Department of Food Science and Technology, National University of Singapore, Science Drive 2, Singapore 117542, Singapore; 3Department of Agriculture and Food Technology, Karakorum International University, Gilgit 15100, Pakistan; 4China-Singapore International Joint Research Institute, Knowledge City, Guangzhou 510663, China

**Keywords:** starch film, packaging, edible, reinforcement, interface, instant noodle

## Abstract

Edible starch-based film was developed for packaging seasoning applied in instant noodles. The edible film can quickly dissolve into hot water so that the seasoning bag can mix in the soup of instant noodles during preparation. To meet the specific requirements of the packaging, such as reasonable high tensile properties, ductility under arid conditions, and low gas permeability, hydroxypropyl cornstarch with various edible additives from food-grade ingredients were applied to enhance the functionality of starch film. In this work, xylose was used as a plasticizer, cellulose crystals were used as a reinforcing agent, and laver was used to decrease gas permeability. The microstructures, interface, and compatibility of various components and film performance were investigated using an optical microscope under polarized light, scanning electron microscope, gas permeability, and tensile testing. The relationship was established between processing methodologies, microstructures, and performances. The results showed that the developed starch-based film have a modulus of 960 MPa, tensile strength of 36 Mpa with 14% elongation, and water vapor permeability less than 5.8 g/m^2^.h under 20% RH condition at room temperature (25 °C), which meets the general requirements of the flavor bag packaging used in instant noodles.

## 1. Introduction

Developing eco-friendly packaging materials is no longer an option, but it has become an urgent necessity since many countries have restricted non-biodegradable materials, particularly for disposable packaging materials. The simplest way to treat the food packaging is to eat them with the foods packaged in. Edible packaging has attracted increasing attention [1,2,3,4], and is mainly based on polysaccharide and protein materials. Starch is the most promising material because of its natural edibility, overwhelming abundance, and annual renewability. Edible starch-based films have been developed and widely used in food and medicine packaging [5,6,7,8], such as applications in candy wrappers and medicine capsules [9,10,11,12].

The bag used for packaging seasonings has been widely used in instant noodles, easy soup, and various pre-prepared ingredients for cooking. It was reported that, on average, everyone ate about 13.6 bags of instant noodles, and the market was more than one myriad bag in 2018 in the world [13]. Traditionally, the films used for flavor bags are bi-orientation polypropylene (BOPP) or polyamide (BOPA) with reasonably excellent mechanical performance and gas barrier properties. Ideally, the packaging film used for the flavor bag would be edible. It could easily dissolve into the hot water during the noodle preparation so that all the treatments before (carefully tearing and pressuring the bag) and after (waste) eating can be omitted, which avoids any environmental issues. An edible starch-based film should be an ideal candidate.

Food and non-food packaging have benefited from starch films’ development [5,10,11,12,14,15]. Improvement of the mechanical properties of starch-based materials is an ongoing challenge due to its poor mechanical performance, particularly tensile strength [16,17,18,19]. Several improvements to the mechanical properties of starch-based materials is an ongoing challenge due to its poor mechanical performance, and tensile strength compositing and blending strategies have been invented to increase these mechanical qualities, such as reinforcing with mineral and natural fillers, or mixing with various decomposable polyesters [20,21,22,23,24,25,26,27,28,29]. However, any additive in edible packaging films is sensitive because of a safety issue or a hazardous risk. All the additives, including plasticizers and reinforcing agents, must be food-grade ingredients.

The aim of this work is to develop edible starch-based film used for packaging seasoning in instant noodles through considering and novelly combining multiple factors, including plasticing, reinforcing, and barriering. The film used for packaging seasonings would keep with the noodles constantly under low humidity conditions (20–30% RH) in a plastic bag or container to maintain the shelf-life of the noodles. The starch-based film needs to be plasticized and reinforced to meet these requirements. Another challenge is to boost the gas barrier function of the film to maintain the flavors. Because it possesses appropriate mechanical and processing capabilities, this investigation used a commercially available food-grade modified (hydroxypropyl) cornstarch (HPCS) as a matrix. It has previously been used to make pharmaceutical capsules and edible films for food packaging [5,11,12]. In order to further improve the performance of the film, multiple additives, including plasticizers, reinforcing agents, and barrier fillers were used, where xylose was used as the predominant plasticizer, cellulose crystals were used as a reinforcing agent, and laver was used to decrease gas permeability. Laver is a very popular seaweed food in Asian countries, such as in Japan, China, Korea, etc. Its popular name in China is Zicai (purple vegetable). Since it has the natural structure of a thin film, the laver can be used as a gas barrier efficiently [5]. The effect of these added ingredients on the starch film functionality was analyzed by an optical microscope under polarized light, electronic scanning microscope, gas permeability, and tensile testing. The multiple relationships among the starch and different additives were studied and used to guide the film development.

## 2. Materials and Methods

### 2.1. Materials

The ingredients utilized in the current study are all readily accessible in the marketplace. Hengrui Starch Company, Luohe, China, provided commercially accessible hydroxypropyl cornstarch (HPCS) (DS 0.4%, moisture content 13 wt%, 23% amylose content). It has good mechanical characteristics [5,11,12]. Food-grade cellulose crystals were procured from Qianrun Bioengineer (Wuhan, China), with a particle size of about 5 mm. Figure 1 shows images of the cellulose crystals under SEM and optical microscope polarized lights. Laver was acquired at a nearby market (Guangzhou, China). The laver, with about 6 mm thickness, was firstly crushed into around 0.5 mm diameter fine particles to distribute it homogeneously in the starch suspension. The laver has a protein content of around 31.3% d.w. Figure 2 depicts images of the laver taken with normal and polarized lights under an optical microscope. Tianjin Kemeou Chemical Reagent Company (Nanjing, China) provided xylose (99.8% pure). Sinopharm Chemical Reagent Company Limited (Shanghai, China) provided 99.5% pure glycerol.

### 2.2. Sample Preparation

The final and optimized starch film contains plasticizer (water, glycerol, and xylose), reinforced cellulose crystals, and laver as the barrier filler. The effect of each additive on the starch-based film was initially studied separately, then mixed according to optimized formulations. The baseline was established as the starch-based film plasticized with water and glycerol. The specimen codes and compositions are included in Table 1. Film thickness and moisture content were measured after the films were kept under 20% RH, and 25 °C temperature.

The starch solutions were made in a beaker using the solution-casting methodology (10% *w*/*w*), in which 10 g (10% *w*/*w*) of starch on a dry basis was dissolved in 90 g water (90% *w*/*w*). Glycerol was added at a constant weight of 0.5 g (5%) (*w*/*w*) in proportion to the 10 g of starch on a dry basis. A referee starch film was made by pre-mixing the solution, then heating it 25 °C to 99 °C, where that was kept for 1 h while continually stirring it with a magnet. The gelatinized starch suspension was rapidly stirred for 45 min before being put onto a polystyrene plate (diameter 10 cm). The film was then dried in an oven for 10–12 h at around 35 degrees Celsius to get a consistent weight. The film containing xylose was prepared by adding concentrations of 5%, 10%, 15%, and 20% (0.5, 1.0, 1.5, and 2.0 g, respectively) into the hot gelatinized suspension of the starch prepared reference sample. The cellulose crystal films were prepared by adding concentrations of 2%, 5%, 7%, and 10% (0.2, 0.5, 0.7, and 1.0 g, respectively) into the hot gelatinized suspension of the starch prepared, the same as the reference sample. The films containing laver were prepared by adding concentrations of 10%, 20%, and 30% (1.0, 2.0, and 3.0 g, respectively) into the hot gelatinized suspension of the starch prepared, the same as the reference sample. Cast films had a final thickness of around 150 µm, and this was measured by estimating how much starch suspension was placed on the plate. A micrometer (μm) was used to determine the actual thickness of the dried film.

### 2.3. Tensile Characteristics

The tensile properties were tested using the standards of ASTM D882-12. The prepared starch films were cut into uniformly-sized tensile bar-shaped specimens. The tensile testing apparatus (Instron 5565) was used at 25 °C with the cross-head at 5 mm/min^−1^ stretching speed. All specimens were conditioned for 48 h before testing in a temperature-humidity box (Lab Companion, Qingdao, China) with a humidity of 20% RH. The data are based on the mean value of the seven samples.

### 2.4. Moisture Effect and Permeability

The moisture effect on the film was tested using contact angle changes with water droplets. The sessile drop methodology was used to analyze water’s liquid/solid/contact angles on sample surfaces using a contact angle goniometer: model OCA 20 from Dataphysics (Dataphysics instrument Co., Filderstadt, Germany). Based on the results of three separate drops, average contact angles were calculated.

As per the ASTM E96/E96M-14 standardized process [30], the water vapor transfer rate (WVTR) was measured in triplicates using a thermos hygrometer and deionized water. Distinct glass cups with a depth of 2.5 cm and a diameter of 4 cm were used to measure the film’s WVTR. A scissor cut the films into a circular form with a length marginally greater than the cup diameter. Each cup was covered with film samples after being filled with anhydrous CaCl_2_. Every cup was placed in a desiccator with a small beaker containing saturated NaCl solution at the surface. To ensure that the saturated solution remained saturated at all times, a small quantity of solid NaCl was left at the bottom. At room temperature, the saturated NaCl solution in the desiccators maintained a constant RH of 75%. Increment in the weight of the cup was used to determine the water vapor transport. To calculate the water vapor transmission rate (WVTR), the slope (g/h) was divided by the transfer area (m^2^).

### 2.5. Characterization of Microstructure and Morphology

#### 2.5.1. Scanning Electron Microscope (SEM)

To examine the film’s surface and the contact between the laver and starch matrix, a scanning electron microscope (SEM, Thermo Fisher Scientific Inc., (NYSE: TMO), Waltham, MA, USA) was utilized. All specimens were gold-coated under vacuum using an Eiko Sputter Coater, and mounted on metal stubs that had been previously coated with double-sided glue. In this experiment, a low voltage of 3 kV was used to minimize the risk of damaging the surface.

#### 2.5.2. Optical Microscope (OM)

The morphology of scattered lavers in the matrix of starch was observed through an optical microscope (Axioskop 40 pol/40 A Pol, ZEISS) linked with a 35 mm SLA camera. XTZ Optical Instrument Factory’s stage micrometer Type C1 slide (1 div = 0.01 mm) was used to calibrate the microscope. Each sample was examined using a microscope at a magnification of 500×, where the photographs were 1.3 times larger than their original size. The crystalline structures of cellulose contained in the laver were studied using polarized light.

### 2.6. Statistical Analysis

All quantitative date were statistically analyzed and presented as means ± standard deviation (SD). The data were analyzed by one way analysis of variance (ANOVA) using SPSS 17.0 package (IBM, Armonk, NY, USA), and when *p* < 0.05, samples were considered to have significant differences according to Duncan’s multiple range tests.

## 3. Results

### 3.1. Effect of Various Additives on Performance of Starch-Based Film

The instant noodles were manufactured in less than 50 RH% conditions, and the water content contained in the noodles itself was less than 5%. The relative humidity in the plastic bag or container is about 20–30% RH. Thus, all the performances of the film were evaluated under 20% RH to meet the conditions. Table 2 lists the effect of various additives on starch-based film, including tensile properties, WVTR, and contact angle. The effect of each additive was initially investigated individually, and then optimized based on the results.

#### 3.1.1. Effect of Cellulose Crystals

The inclusion of the cellulose crystals considerably enhanced the starch film’s strength, as depicted in Table 2. After the crystals were added, both modulus and tensile strength rose. Because hard crystal particles worked as reinforcing agents, this was predicted. The reinforcing process of stiff particles in a matrix is widely understood, and crystals behave similarly. The modulus and tensile strength of the crystals steadily rose as the crystal content increased. On the other hand, the results demonstrate that adding crystals to the films has a minor effect on the elongation at the break. It has to be pointed out that both starch film and the film reinforced by the cellulose crystals are generally very brittle since the elongations are very low, with a value of only about 3–4%. Because of this brittle behavior, the standard dividends (±) of the elongation is higher (>20%).

The starch film WVTP was not affected by the cellulose crystals. It was not expected that a rigid particle could reduce the gas permeability if it has much low gas permeability, since the gap or weak interface between the particle and matrix significantly increases gas permeability. The results indicate that the interface between the starch and cellulose particle is excellent. CA of the starch film improved marginally after adding the cellulose crystals, indicating the moisture sensitivity of the film decreased. 

#### 3.1.2. Effect of Laver

The modulus elevated when adding more laver, and steadily increased when enhancing laver concentration, indicating that laver improves the films’ stiffness as predicted. The tensile strength rose slightly as well, especially when the laver loading was more than 20%. One theory is that the stiff fibers strengthen the coatings in the laver cell walls. As with how most other fillers behaved, elongations of starch-based films reduced marginally with laver content, and steadily declined when increasing laver content. Similar to the reinforcement by the cellulose crystals, the elongation standard dividends (±) of the starch-based film comprising laver are also reasonably high.

CA increased as laver content increased, implying that the starch-based film’s moisture sensitivity diminished. The findings explain why the laver is moisture resistant. The laver’s moisture resistance is due to its semi-crystalline cellulose and protein [5].

After adding laver to starch-based films, the WVTP declined significantly, and the WVTP decreased even more when the laver content was raised. Creating a channel with some twists in the film matrix that help water vapors go through reduces WVTP when laver is induced [5]. Furthermore, creating the inter-link between protein and starch [30] may lower gas permeability because the tight interlinking between protein and starch has a lower free volume, and allows for less diffusion of moisture. Edible films produced by a combination of whey protein and starch have shown similar results [31,32].

#### 3.1.3. Effect of Xylose

The mechanical performance of starch-based materials relies on plasticizer content (water or others). To achieve reasonable flexibility (toughness) under arid conditions (20% RH), different plasticizers were used in this work. The most common plasticizer system used for starch-comprised products is water with glycerol, since glycerol has assertive moisture absorption behavior and is used to maintain the water in the materials. However, the water is still not stable under low RH conditions. A constant concentration of water/glycerol 10/3 was used in this work, and additional plasticizer xylose from food ingredients was added. Xylose is a monosaccharide with a linear structure which has shown remarkably high efficiency in destroying the order structures, and enhancing the movement of polymer chains. As a result, plasticization efficiency improves. [31] Table 2 displays that xylose considerably enhanced the toughness of the starch film, with elongation increasing by up to eight times. With more xylose, the elongation rate increased. However, when the xylose level increased, the modulus and tensile strength of the material declined.

WVTR slightly decreased after the additional xylose. CA was slightly enhanced by xylose, indicating that the hydrophobicity of the starch film marginally increased. The hydroxyl groups in linear xylose are more flexible and face out, which makes sense [33].

### 3.2. Morphologies and Interface

#### 3.2.1. Surface Morphologies

The linkage between cellulose crystals or laver and the starch matrix was investigated using SEM (see Figure 3). It is visible that the pure starch film has a relatively smooth surface (Figure 3A). The cellulose crystals and laver can be seen on the film’s surface after the film’s dried and shrinking matrix. The interface between the starch matrix and cellulose crystalline (B) or laver (C) was homogeneous, and there was no space between the particles and the matrix, which implies that polysaccharide crystals are fully compatible inside the starch matrix. The mechanism may be applied to describe why the films have better mechanical characteristics.

#### 3.2.2. Interface

The cross-section images of the films using SEM were used to investigate the linkage between the fillers and the starch matrix (see Figure 4). The pure starch film’s cross-section picture has a smooth surface (a). Even after being sliced, the crystals and laver may still be detected in the starch matrix (b). At higher temperatures, the laver’s cell structure has been disrupted and expanded. These findings were in accordance with the surface morphology investigation (Figure 4), and may be utilized to elaborate mechanical characteristics and moisture barriers. The thin laver also formed a twisted gas channel in the film matrix, which reduced gas permeability.

### 3.3. Developing Flavor Bag Packaging

The above results helped to develop coating, and extrude the starch-based films and various flavor bag packagings. Figure 5 depicts the images of extruded starch-comprised film and flavor bag packaging without laver (A,B), since some industries require high gas barriers and others do not. The films were developed based on the CX and CLX films presented in Table 1, respectively. All the starch-based flavor bags dissolve in hot water (85 °C) within a few seconds.

## 4. Discussion

It is well established that the functionality of starch-based products strongly depends on plasticizers and environmental conditions. One of the key challenges of applying starch-comprised film in flavor bag packaging is keeping the film with a good strength and toughness. Improving gas permeability is an additional advantage. In this work, xylose was used as a plasticizer, cellulose crystals were used as a reinforcing agent, and laver was used to decrease gas permeability. Tensile strength improved, according to the findings, after the addition of both cellulose crystals and laver. The laver also decreased WVTR significantly. But the flexibility or toughness of the starch-based film became worse. The films are generally brittle since they have very low elongation at the broken point with or without cellulose crystals and laver under low RH conditions. Additional xylose significantly improved the toughness, and the elongation increased up to 7–8 times. The mechanical properties could be balanced through designing and optimizing various additives. Moisture sensitivity also decreased after adding cellulose crystals and laver, which is crucial for food packaging.

The morphologies of the starch-based film have been investigated using SEM to explain the results. Both surface morphologies and cross-section images showed that starch matrix was compatible with cellulose crystals and laver. That was an excellent indication of a linkage between the strengthening components and the matrix. This was expected, as both starch and cellulose contained in crystals and laver have the same chemical unit, glucose. Actually, that is why they were selected as suitable materials for this work.

## 5. Conclusions

The edible starch-based film has been successfully developed for packaging seasonings applied in instant noodles. To meet the specific requirements of the packaging, such as reasonable high tensile properties, ductility under arid conditions, and low gas permeability, several edible additives from food-grade ingredients were applied to enhance the functionality of the starch film. Tensile strength increased after additional cellulose crystals and laver, whereas toughness significantly improved with additional xylose. The laver also decreased WVTR significantly. Based on the particular function of each additive, the balanced and optimized starch-based film for flavor bag packaging has been successfully developed.

## Figures and Tables

**Figure 1 foods-10-03105-f001:**
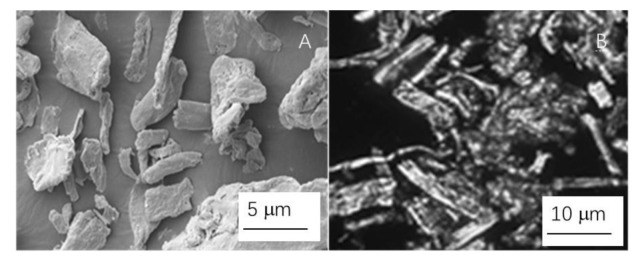
Cellulose crystals under SEM (**A**) and optical microscope polarized lights (**B**).

**Figure 2 foods-10-03105-f002:**
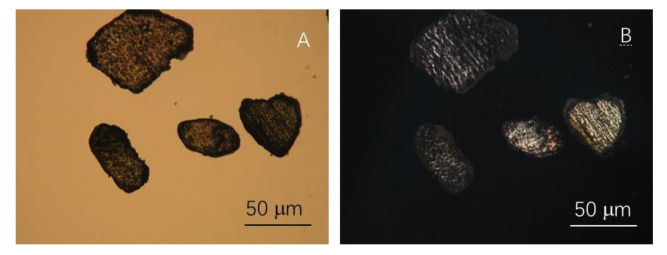
Images of the laver taken with natural (**A**) and polarized (**B**) lights under an optical microscope.

**Figure 3 foods-10-03105-f003:**
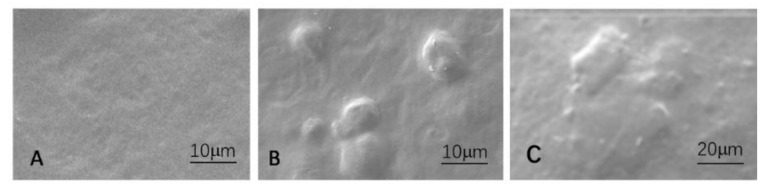
The surface images using SEM. (**A**) The pure starch film, (**B**) film containing cellulose crystals, (**C**) film with the laver.

**Figure 4 foods-10-03105-f004:**
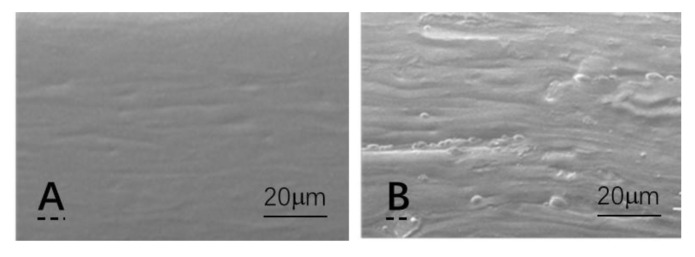
Image of the pure starch film in cross-section using SEM (**A**). The film containing cellulose crystals and laver (**B**) (sample CLX).

**Figure 5 foods-10-03105-f005:**
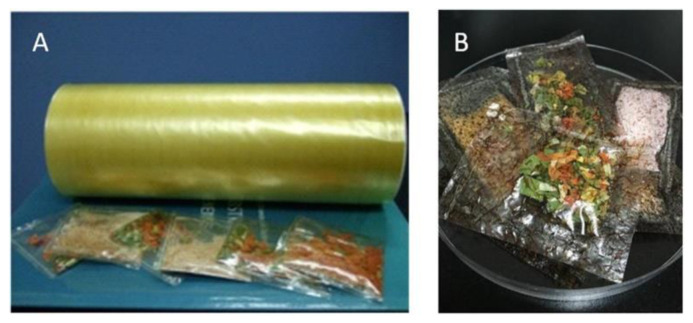
Photos of starch-based film and flavor bag packaging without (**A**,**B**) laver.

**Table 1 foods-10-03105-t001:** Sample code, formulations, as well as film thickness and moisture content.

Sample Code	Cellulose Cryst (*w*/*w*)	Laver (*w*/*w*)	Xylose (*w*/*w*)	Thickness (mm)	Moisture Content (%) ^2^
Starch Film ^1^	-	-	-	146 ± 4	7.01
C-1	2	-	-	160 ± 8	8.62
C-2	5	-	-	171 ± 9	8.53
**C-3**	**7**	-	-	**173 ± 8**	**8.73**
C-4	10	-	-	181 ± 9	8.87
L-1	-	10	-	167 ± 7	9.37
**L-2**	-	**20**	-	**171 ± 6**	**9.77**
L-3	-	30	-	177 ± 8	9.91
X-1	-	-	5	152 ± 8	9.67
X-2	-	-	10	168 ± 6	10.17
**X-3**	-	-	**15**	**167 ± 7**	**10.41**
X-4	-	-	20	165 ± 6	10.85
**CX film**	**7**		**15**	**171 ± 7**	**9.71**
**CLX film**	**7**	**20**	**15**	**178 ± 9**	**9.55**

Notice: ^1.^ All of the mentioned additives are made up of 0.5% glycerol (0.5 g), 10% dried starch (10 g), and 90% water (90 g). The ratio to starch (*w*/*w*). ^2.^ The films were stored at a temperature of 25 °C, and a humidity of 20%. Sample with C- means containing cellulose, L- means containing laver, X- means containg xylose. Bold means these formulations were used as baseline for CX and CLX.

**Table 2 foods-10-03105-t002:** Effect of various ingredients on the starch film mechanical performances, WVTR, and contact angle.

Sample Code	Modulus (MPa)	Tensile Str (MPa)	Elongation (%)	WVTR (g/(m^2^ h))	Contact Angle (Ɵ)
Starch Film	1352 ± 119 ^a^	45.3 ± 2.9 ^a^	3.9 ± 0.7 ^b^	16.2 ± 3.4 ^a^	82.0 ± 2.9 ^c^
C-1	1387 ± 102 ^b^	45.5 ± 4.8 ^b^	3.7 ± 0.7 ^a^	16.0 ± 2.8 ^ab^	83.6 ± 2.7 ^b^
C-2	1429 ± 118 ^ab^	47.2 ± 4.7 ^ab^	3.4 ± 1.1 ^ab^	15.6 ± 1.9 ^ab^	85.5 ± 3.3 ^ab^
C-3	1492 ± 72 ^a^	48.2 ± 5.2 ^ab^	3.3 ± 0.8 ^ab^	15.3 ± 3.5 ^b^	85.7 ± 3.6 ^ab^
C-4	1533 ± 126 ^a^	51.4 ± 5.6 ^a^	2.6 ± 0.6 ^b^	16.1 ± 2.7 ^a^	87.9 ± 4.1 ^a^
L-1	1382 ± 122 ^b^	46.8 ± 3.9 ^b^	3.4 ± 2.1 ^a^	6.7 ± 0.5 ^a^	91.4 ± 3.6 ^a^
L-2	1410 ± 116 ^ab^	47.7 ± 4.8 ^ab^	3.1 ± 1.6 ^ab^	5.7 ± 0.4 ^b^	105.8 ± 4.2 ^b^
L-3	1432 ± 104 ^a^	51.1 ± 5.9 ^a^	2.7 ± 0.2 ^b^	5.3 ± 0.6 ^b^	119.9 ± 3.8 ^c^
X-1	1242 ± 79 ^a^	42.6 ± 3.2 ^a^	8.6 ± 2.2 ^d^	13.2 ± 0.8 ^b^	87.7 ± 3.9 ^a^
X-2	967 ± 74 ^b^	36.1 ± 3.1 ^b^	11.3 ± 1.5 ^c^	14.8 ± 1.3 ^ab^	94.2 ± 3.4 ^a^
X-3	818 ± 81 ^c^	27.0 ± 2.3 ^c^	19.9 ± 2.1 ^b^	15.7 ± 1.1 ^a^	102.4 ± 2.8 ^b^
X-4	620 ± 66 ^d^	21.3 ± 1.7 ^d^	28.4 ± 2.3 ^a^	15.5 ± 1.6 ^a^	103.9 ± 2.7 ^c^
CX film	925 ± 73 ^b^	37 ± 3.6 ^b^	16.1 ± 2.6 ^a^	15.6 ± 0.5 ^a^	91.8 ± 2.7 ^b^
CLX film	965 ± 81 ^b^	36 ± 2.3 ^b^	14.2 ± 2.1 ^a^	5.8 ± 0.4 ^b^	97.9 ± 2.9 ^a^

The films were kept at room temperature under 20% RH. The data were analyzed by one way analysis of variance (ANOVA) and marked as ^abcd^.

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
