# Peer review of "Developing Edible Starch Film Used for Packaging Seasonings in Instant Noodles"

_foods, 2021, doi:10.3390/foods10123105_

Round 1

Reviewer 1 Report

See attached file for detailed comments.

General remarks:

  • Title: The word “flavor” is present in the title, but the flavor aspect is not studied in particular in the manuscript and therefore should not appear.
  • Hydroxypropyl-cornstarch should be defined as acronym (HPCS for example) in the beginning and used instead of “starch” in the rest of the text to avoid the confusion with non-chemically modified native granular or destructurised starch.
  • You should use only one acronym WVTR or WVP for water vapour porosity in the whole manuscript.

Author Response

  • Title: The word “flavor” is present in the title, but the flavor aspect is not studied in particular in the manuscript and therefore should not appear.

Response: We agree with this suggestion and remove it in the title. 

  • Hydroxypropyl-cornstarch should be defined as acronym (HPCS for example) in the beginning and used instead of “starch” in the rest of the text to avoid the confusion with non-chemically modified native granular or destructurised starch.

Response: We agree with this suggestion and changed it. 

  • You should use only one acronym WVTR or WVP for water vapour porosity in the whole manuscript.

Response: We agree with this suggestion and changed it. 

Reviewer 2 Report

This manuscript presents the effect of crystal cellulose, laver, and xylose as additives on the edible starch-based films properties. The research design is appropriate, the methods are satisfactorily described, and the results are clearly presented. In my opinion, it only needs minor revision.

Minor comments

In Table 2: The values on contact angle should be given with one decimal digit (according to standard deviation)

The experimental results should be processed by statistical analysis (ANOVA) and the statistically significant different values should be pointed out.

Author Response

We have corrected the Table-2 and provided one decimal digit for CA. 

Reviewer 3 Report

In this paper developing edible starch-based film for flavor bag packaging
used in instant noodles were investigated. Reviewed work is interesting from the scientific point of view and covers important considerations regarding biodegradable materials. However, the manuscript should be improved. 

  1. The introduction section must be improved.
  2. Line 50: The sentence “Starch-based materials ' mechanical characteristics are consistent because of their low mechanical characteristics, specifically tensile strength” should be rewritten.
  3. The aim and the novelty of the paper should be highlight. It is necessary to discuss what additives will be add to starch and why.What are their unique properties that can make the obtained products meet the assumed goals? Why did the authors use such and not other raw materials? Is the use of these raw materials profitable on a larger scale?
  4. Line 76: The biological material from which the cellulose was obtained was not described. “Food-grade cellulose” – please specify.
  5. Line 79: “Laver was acquired at a nearby market (Guangzhou, China)”. Is there any other name for this raw material? I have not met in this kind of material? Is it edible seaweed? There is no description of the material used in the introduction
  6. Line 82: “DW” maybe better d.w.
  7. Line 99: film thickness and moisture content should be described in methods section.
  8. 1. Please explain the abbreviations. What does mean X and CX film and CLX film. It was not explain in the text. The description of the methodology does not explain this exactly.
  9. Line 127: Please provide the length and width of speciment.
  10. Line 215: Was xylose only added to pure starch or was in cellulose and laver samples? the methodology does not specify this
  11. Line 215: There have been many papers describing the effect of the addition of various plasticizers and their concentration to starch.
  12. Line 207: The results should be supported by a more detailed discussion. Also based on literature data. The current discussion does not add much.

Recommendations - Major Revisions for reconsideration
The paper is interesting from the scientific point of view. There were used proper methods to gain the aim. However, the manuscript have many limitations that must be corrected.

Author Response

  1. The introduction section must be improved.

Response: We have modified it.

  1. Line 50: The sentence “Starch-based materials ' mechanical characteristics are consistent because of their low mechanical characteristics, specifically tensile strength” should be rewritten.

Response: We agree with this suggestion and rewrite it.

  1. The aim and the novelty of the paper should be highlight. It is necessary to discuss what additives will be add to starch and why. What are their unique properties that can make the obtained products meet the assumed goals? Why did the authors use such and not other raw materials? Is the use of these raw materials profitable on a larger scale?

Response: We agree with this suggestion and modified this section. For example, 1) all the additives are added to improve the mechanical and gas permeability behaviors of the starch film. 2)the unique properties of all the additives are edible.

  1. Line 76: The biological material from which the cellulose was obtained was not described. “Food-grade cellulose” – please specify.

Response: We have added it (Qianrun Bioengineer (Wuhan, China)).

  1. Line 79: “Laver was acquired at a nearby market (Guangzhou, China)”. Is there any other name for this raw material? I have not met in this kind of material? Is it edible seaweed? There is no description of the material used in the introduction

Response: It is a very popular seaweed as food in Asian countries, such as Japan, China and Korean etc. Its popular name in China is Zicai (Purple veritable).

  1. Line 82: “DW” maybe better d.w.

Response: We have changed it.

  1. Line 99: film thickness and moisture content should be described in methods section。

Response: We fully understand this suggestion and consider they should be in Table-2. Since both thickness and moisture content were controlled by the preparation (experimental work) and there are no room in Table-2, we list them in Table-1.  

  1. Table-1. Please explain the abbreviations. What does mean X and CX film and CLX film. It was not explain in the text. The description of the methodology does not explain this exactly.

Response: We have added the explanation.

  1. Line 127: Please provide the length and width of specimen.

Response: We have added them (polystyrene plate (diameter 10cm)).

  1. Line 215: Was xylose only added to pure starch or was in cellulose and laver samples? the methodology does not specify this

Response: We have added this information in Table-1.

  1. Line 215: There have been many papers describing the effect of the addition of various plasticizers and their concentration to starch.

Response: We agree with this comment. The most popular ones are glycerol and xylose since they are edible. Both of them have been used in this work. Since glycerol has strong behavior of moisture absorption, xylose was also used to balance them.  

  1. Line 207: The results should be supported by a more detailed discussion. Also based on literature data. The current discussion does not add much.

Response: We agree with this suggestion and added more ref to support it.

Reviewer 4 Report

The article “Developing edible starch-based film for flavour bag packaging used in instant noodles” by Hui Chen et al. is structured following the classic model for this type of material (Research Article), comprising four parts: Introduction, Materials and Methods, Results and Discussion, and Conclusions. The four major components of the article are presented coherently and logically, tightly linked to one another. The list of bibliographic references is adequate; the documentation is appropriate regarding the titles consulted.

In my opinion, the article presented is quite good. The introduction of the manuscript, in general, is good. The same applies to the materials and methods. As far as the discussion is concerned is very clear and understandable. All the work developed in the manuscript is described and presented in a very understandable way, is translated into a logical text, and easy to understand the relationship between the different existing parts. In my opinion, this work is undoubted progress in the studied subject.

Although the article, in general, is quite good, I think the authors should add one question in the introduction, which is quite pertinent and would raise the quality of the submitted manuscript. In my opinion, there will always have to be a package in which to place these new packages. As these packages are edible, they certainly cannot be placed on the market without being protected from direct contact with the outside, at the risk of putting into question their good condition for consumption. Therefore, what is the real benefit of these packaging’s, since the option made for external packaging may not be in line to guarantee the sustainability of the entire process. It would be interesting to see this discussion expressed in the submitted manuscript.

Author Response

Although the article, in general, is quite good, I think the authors should add one question in the introduction, which is quite pertinent and would raise the quality of the submitted manuscript. In my opinion, there will always have to be a package in which to place these new packages. As these packages are edible, they certainly cannot be placed on the market without being protected from direct contact with the outside, at the risk of putting into question their good condition for consumption. Therefore, what is the real benefit of these packaging’s, since the option made for external packaging may not be in line to guarantee the sustainability of the entire process. It would be interesting to see this discussion expressed in the submitted manuscript.

Response: Thanks for this comment and suggestion. It is a very interesting suggestion. Our response so far is limited in “the issue of external packaging” need to be solved by different ways, such as biodegradable polyesters (modified PLA or PBAT). As the reviewer mentioned the edible packaging film “cannot be placed on the market without being protected from direct contact with the outside”. The benefit should be 1) reducing packaging waster; 2) easily to prepare the noodle (not necessary to tear the seasoning bags).

Round 2

Reviewer 3 Report

Dear Authors,

I still do not see an improvement in the work in terms of the proposed suggestions. Applies to: 

  1. The aim and the novelty of the paper should be highlight. It is necessary to discuss what additives will be add to starch and why. What are their unique properties that can make the obtained products meet the assumed goals? Why did the authors use such and not other raw materials? Is the use of these raw materials profitable on a larger scale? The introduction section must be improved,
  2. "It is a very popular seaweed as food in Asian countries, such as Japan, China and Korean etc. Its popular name in China is Zicai (Purple veritable)" Please add this in material section and described in introduction section.
  3. The introduction section must be improved. Please explain why the use of  laver is so profitable.

Author Response

  • The aim and the novelty of the paper should be highlight. It is necessary to discuss what additives will be add to starch and why. What are their unique properties that can make the obtained products meet the assumed goals? Why did the authors use such and not other raw materials? Is the use of these raw materials profitable on a larger scale? The introduction section must be improved,

Response: We have modified Introduction section according to the suggestion.” All the additives used in this work are based on literature research and our previous work (mentioned in introduction, results and discussions).

  • "It is a very popular seaweed as food in Asian countries, such as Japan, China and Korean etc. Its popular name in China is Zicai (Purple veritable)" Please add this in material section and described in introduction section.

Response: We have added this in Introduction section according to the suggestion.

  • The introduction section must be improved. Please explain why the use of laver is so profitable.

Response: We have added this in Introduction section according to the suggestion.